# INSNET: An Efficient, Flexible, and Performant Insertion-based Text Generation Model

**Sidi Lu, Tao Meng, Nanyun Peng**
University of California, Los Angeles
{sidilu, tmeng, violetpeng}@cs.ucla.edu

## Abstract

We propose INSNET, an expressive insertion-based text generator with efficient training and flexible decoding (parallel or sequential). Unlike most existing insertion-based text generation works that require re-encoding of the context after each insertion operation and thus are inefficient to train, INSNET only requires one pass of context encoding for the entire sequence during training by introducing a novel insertion-oriented position encoding and a light-weighted slot representation strategy to enable computation sharing. Furthermore, we propose an algorithm INSNET-Dinic to better determine the parallelization of insertion operations that provides a controllable switch between parallel and sequential decoding, making it flexible to handle more parallelizable tasks such as machine translation with efficient decoding, or less parallelizable tasks such as open-domain text generation to guarantee high-quality outputs. Experiments on two lexically constrained text generation datasets and three machine translation datasets demonstrate IN-SNET's advantages over previous insertion-based methods in terms of training speed, inference efficiency, and generation quality.

## 1 Introduction

Insertion-based text generation that formulates the generation process as a sequence of token insertion operations has received increasing attention in recent years. There are two major advantages of insertion-based generation over the prevalent left-to-right auto-regressive paradigm: 1) It reduces the decoding cost to sub-linear *w.r.t.* the sequence length with parallel decoding (Stern et al., 2019; Gu et al., 2019b), and 2) the flexible insertion orders may better recover/utilize the underlying linguistic structures of languages (Welleck et al., 2019; Gu et al., 2019a).

However, this new paradigm of text generation brings unique challenges, mostly in the training efficiency. Unlike left-to-right auto-regressive decoders which monotonically expand the context, the insertion operations complicate the position information of each token as the context expands. Concretely, as is shown in Figure 1, the absolute position of a token in a sequence constantly changes along with the insertion operations. As a result, a naive implementation of insertion-based models (e.g., Stern et al. (2019); Gu et al. (2019b)) needs to re-encode the context with updated positional information for each token as the insertions proceed, yielding inefficient training with $O(n)$ times of context re-encoding (with $n$ indicating the sequence length).

To tackle this problem, previous insertion-based generation models such as Insertion Transformer (InsT) (Stern et al., 2019) and Levenshtein Transformer (LevT) (Gu et al., 2019b) propose parallel token insertion to reduce the insertion/re-encoding steps from $O(n)$ to $\Theta(\log n)$ for both training and inference. However, while it works well for machine translation, such parallel insertion falls short on high-entropy generation tasks such as open-domain dialogue systems(Li et al., 2017a), creative generation such as stories (Yao et al., 2019; Goldfarb-Tarrant et al., 2020; Han et al., 2022), poetry (Manurung et al., 2000; Tian and Peng, 2022), and humor generation (Hempelmann, 2008; He et al.,

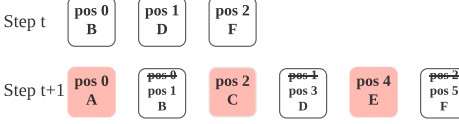

Figure 1: The absolute position information of each token is volatile for insertion-based models. Thus, naive models with absolute position encoding have to re-encode the sequence after each insertion operation during training.

| Model | #Re-Enc. w/ ParaDec | #Re-Enc. w/ SeqDec | PosInfo |
|---|---|---|---|
| Ins. Trans. | $\Theta(\log n)$ | $O(n)$ | Absolute |
| Lev. Trans. | $\Theta(\log n)$ | $\Omega(n)$ | Absolute |
| NMSTG | $O(n)$ | $O(n)$ | Markovian/Absolute |
| InDIGO | N/A | $O(1)$ | Direction-only |
| INSNET (Ours) | $O(1)$ | $O(1)$ | Relative |

Table 1: Comparisons between INSNET and existing insertion-based models regarding training-time re-encoding steps (*#Re-Enc.* columns) and how the models encode positional information (the *PosInfo* column).

2019; Mittal et al., 2022; Tian et al., 2022; Sun et al., 2022), where the output is open-ended and the tokens are usually highly dependent on each other, thus cannot be inserted in parallel. Another thread of work seeks to maintain sequential insertions but simplify the position encoding for the tokens so that they do not change as the insertion operations proceed (Gu et al., 2019a) to save computation. However, this sacrifices model capability and limits its applications to complex sequences.

In this paper, we propose INSNET, which addresses the training efficiency issue of insertion-based text generators by enabling computation sharing similar as that in vanilla decoder transformers. **Our first contribution** is an insertion-oriented, relative positional encoding coined *offset* that allows INSNET to achieve computation sharing without compromising model capacity. At each insertion step, previously computed position encodings of the existing tokens remain unchanged, while the position encodings of newly inserted tokens accurately recording the updated pairwise token spatial relations of all the inserted tokens. A corresponding offset can be efficiently computed for any given insertion order with a novel process named *offset compression*. In this way, we avoid expensive re-encoding of the context, and based on the encoded context, **our second contribution** is the design of an effective aggregation strategy that allows us to parallelly generate expressive slot representations for every slot in the partially generated sequence, flexibly support both sequential and parallel decoding. **Finally**, inspired by the layerization idea of Dinic's algorithm(Dinitz, 2006), we propose an algorithm for INSNET to better determine the parallelization of insertion operations in order to reduce the likelihood discrepancy before and after the parallelization. With all the components, INSNET is a novel framework that enables efficient training, flexible decoding, and expressive positional encoding. Table 1 shows a comparison between INSNET and all prior insertion-based models.

## 2 Related Works and Background

**Auto-regressive Language Models** minimize the negative-log-likelihood of a sequence of n tokens $\boldsymbol{s}_{<n} = [x_0, x_1, ..., x_{n-1}]$ with a left-to-right factorization. With the transformer architecture (Vaswani et al., 2017), each step of likelihood estimation can be calculated in parallel while sharing the prefix context encoding calculations. This makes it possible to build powerful and efficient text generation models, like the GPT family (Radford et al., 2018, 2019; Brown et al., 2020). A lot of successful applications are based on this paradigm of models, such as automatic story generation (Yao et al., 2019; Tan et al., 2020), image captioning (Vinyals et al., 2015; Xu et al., 2015), machine translation (Bahdanau et al., 2015; Liu et al., 2020), and dialogue system (Li et al., 2017b,a).

**Insertion-based Models with Parallel Decoding** Insertion transformer (Stern et al., 2019) (InsT) proposes a design for insertion-based text generation. In each step, a bi-directional encoder transformer is performed on the partial sequence to compute the representation for each candidate slot between every two consecutive positions. Then, a model for the joint distribution of position-token is built to insert one or more token(s). Variants of InsT share the common atomic objective that models the *step log-likelihood*. On step $t$ where a token $x_{i\downarrow i+1}$ is inserted in slot $\boldsymbol{l}_{i\downarrow i+1}$ between position $i$ and $i + 1$ of context $\boldsymbol{s}_t^o$, the *step log-likelihood* can be written as:

$$\log p(x_{i\downarrow i+1}, \boldsymbol{l}_{i\downarrow i+1}|\boldsymbol{s}_t^o) = \log p_{position}(i + 1|\boldsymbol{s}_t^o) + \log p_{token}(x_{i\downarrow i+1}|e(\boldsymbol{s}_t^o)_i \oplus e(\boldsymbol{s}_t^o)_{i+1}),$$

where $e(\cdot)_i$ stands for the i-th position of bi-directional encoding of the sequence and $\oplus$ stands for vector concatenation. InsT adopts the original absolute positional encoding of transformers, the

representation of the generated sequence has to be *completely re-encoded* after each step of context expansion to match the position changes of tokens. The expectation of the negative log step likelihood over all permitted context-insertion pairs at each step is computed as the *step loss*. The step losses from the first step to the last one are summed up as the *sequence loss*. Benefiting from the partially parallelized prediction of tokens, InsT can reduce the number of re-encoding steps to $\Theta(\log n)$.

Levenshtein Transformer (Gu et al., 2019b) (LevT) contains two phases during generation: 1) Insertion phase: it first uses a similar strategy as InsT, but only inserts placeholders; an MLM is applied to fulfill the placeholders. 2) Deletion phase: the model is trained as a token-wise discriminator to determine where to delete, under the evaluation of the Levenshtein distance by dynamic programming. In practice, this would result in a slower process compared to InsT, but better generation quality as it alleviates the incoherence caused by parallel insertions.

**Sequential Insertion-based Model** Previous exploration in insertion-based models mostly assumed the usage of parallel decoding in each decoding step, resulting in a partially auto-regressive procedure. For those are trained to only generate one token per decoding step, a vanilla implementation would result in an $O(n)$ factor in training time complexity. NMSTG (Non-monotonic Sequential Text Generation, Welleck et al. (2019)) is one of the first attempts at modeling a non-monotonic sequential insertion-based generation process. It constrains the dependencies of each inserted token to pseudo-Markovian on an expansion tree. InDIGO(Gu et al., 2019a) is proposed as a sequentially-decoded insertion-based model with the transformer architecture. It supports the efficient likelihood estimation by working around the aforementioned *volatility problem* (see Figure 1) at a cost of omitting the distance information in its position encoding, and is thus able to adopt the conventional computation sharing trick to boost the multiplicative factor in training time complexity for each sequence to as fast as $O(1)$. However, InDIGO uses a encoding of all previously inserted tokens to predict the next token regardless of their tentative position. It is only after the next token is predicted, the inserted position for it is predicted. Thus, there's no trivial solution to use InDIGO as a (partially) non-autoregressive insertion-based generator.

**Absolute vs. Relative Position Embedding** The original transformer (Vaswani et al., 2017) uses sinusoidal, absolute position embeddings. Relative position embedding in transformers (Shaw et al., 2018; Dai et al., 2019) was originally proposed to make the modeling of spatial relation invariant to position translation, and to improve the long-term dependency performance of the model. In replacement of absolute positions, which are tied to each token in the sequence, relative positions try to encode the spatial layout of the tokens with a matrix that records a directed distance from the column token to the row token. We further develop the idea of relative position embedding as the key component to overcome the issue of volatile position information of absolute positions in an insertion-based generation process. A noticeable fact is that InDIGO's implementation of position encoding can also be regarded as a direction-only relative position embedding system.

# 3 The INSNET Model

There are three major components of INSNET: 1) An context encoder based on the transformer architecture (Vaswani et al., 2017) that uses a novel way to compute *insertion-oriented relative position encoding* to better suit the insertion-based nature of the generation process and enable computation sharing (Section 3.1); 2) a module to compose expressive *slot representation* for predicting tokens to insert in different slots simultaneously (Section 3.2); and 3) an algorithm that adaptively determines the parallelization of the insertion operations to minimize conflicts (Section 3.3). Figure 2 illustrate the components of the full model.

## 3.1 Context Encoder: A Transformer with Insertion-Oriented Relative Position

In the introduction, we discussed the challenges in (efficient) computation sharing caused by the volatile position information of the context tokens as shown in Figure 1. We address this problem by empowering the transformer-based context encoder with a distance-aware, insertion-oriented pairwise relative position encoding. The proposed insertion-oriented relative position encoding shares important designs with previous relative position embeddings(Dai et al., 2019; Yang et al., 2019; Shih et al., 2019), but differs in how it accurately depicts the spatial layout (i.e., the actual sequential order) of the inserted tokens in an insertion-based generation process.

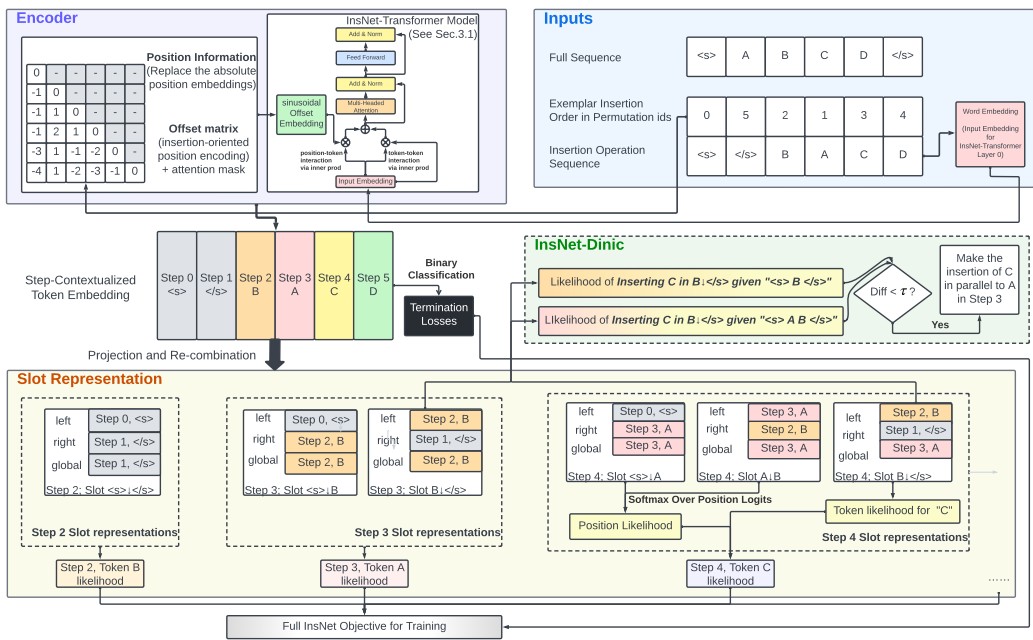

Figure 2: Illustration of the full InsNet model. Given an input sequence with a specified insertion order as illustrated in the [Inputs] panel (the upper right panel in blue), a context encoder (Section 3.1) illustrated in the [Encoder] panel (the upper left corner in purple) with insertion-oriented position encoding (Section 3.1.1) is applied to the sequence of insertion operations to obtain step-contextualized token embeddings as the transformer outputs. Then, the slot representation module (Section 3.2) illustrated in the [Slot Representation] panel (the bottom panel in yellow) compose slot representations for each step from InsNet transformer outputs using its left, right token representation and a global, step-wise token representation. The slot representations are then used to compute the token/position likelihood and also to determine the auto-parallelization in InsNet-Dinic (Section 3.3) illustrated in the [InsNet-Dinic] panel (the middle right panel in green).

### 3.1.1 Offset: Insertion-Oriented Relative Position Encoding

To illustrate our relative position encoding, considering the partial context (to be completed by further insertions) of *"I have pen"* with an insertion order of *"have pen I"*. To insert an *"a"* in between *"have"* and *"pen"*, the distance vector for token *"a"* against the rest of the context inserted so far should be [("have", -1), ("pen", +1), ("I", -2)], simplified as [-1, +1, -2]. This relative position encoding clearly defines where the insertion happens by only describing the pairwise spatial relationship between the incoming token and the existing context tokens. We can pack the relative position vectors for each insertion step to get a matrix that reflects the relative spatial relation along the trajectory of insertions, with each row corresponding to an insertion step. We name it the *offset* matrix.

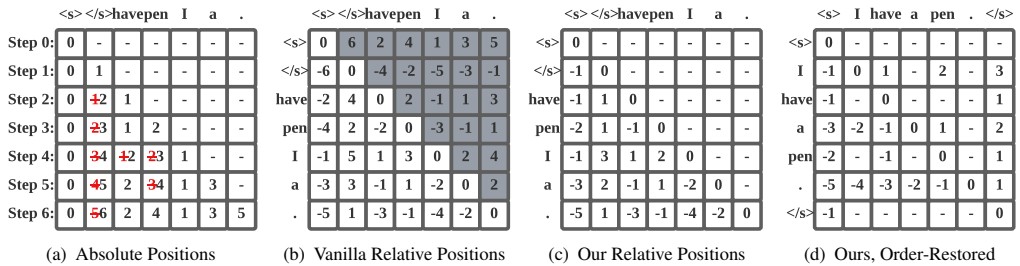

Figure 3: Comparison of different position encodings for an insertion-based generation process. From left to right, we illustrate 3(a) the volatile absolute positions, 3(b) the traditional non-insertion-oriented relative positions, 3(c) the proposed offset matrix presented in the insertion order, and 3(d) the same offset matrix permuted to show how it looks like if we restore the actual (partial-)sequence.

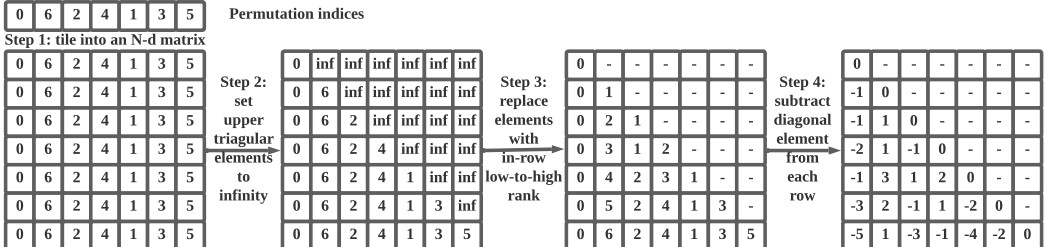

Figure 4: Illustration of the offset compression algorithm. It efficiently transforms the insertion order/permutation indices to an offset matrix.

As a concrete example, to generate the sentence *"I have a pen."* with an insertion order "⟨BOS⟩ ⟨EOS⟩ have pen I a .", the complete offset matrix is shown in Figure 3(c). The effective elements of this matrix are all in the lower triangular region, making the application of the computation sharing trick for the decoder transformers trivial. This reduces the number of context (re-)encoding steps from $n$ in naive insertion-based models (Stern et al., 2019) to 1 during training.

**Discussion: Comparisons With Other Positional Encoding Strategies** To illustrate how the proposed insertion-oriented relative position encoding is different from existing position encoding strategies, Figure 3 shows the volatile, un-reusable absolute positions in 3(a), the traditional non-insertion-oriented relative positions as used in models such as XLNet Yang et al. (2019) in 3(b), the proposed offset matrix in the insertion order in 3(c), and the same offset matrix permuted to show the restored actual (partial-)sequence in 3(d). We can see that the absolute and the traditional relative positional encoding strategies are not applicable for efficient insertion-based generators. The former requires constant updates of the positional encoding of the existing tokens as the insertion progresses, disabling computation sharing. The latter requires knowledge about the final sequence length at the beginning of the insertion process, which is unrealistic and destroys the flexibility of insertion-based models. Our relative position encoding, on the other hand, reflects the order of the original sequence. Tokens that are on the left of the current one in the original sequence have a negative relative position to the current one while preserving the insertion order. As is illustrated in Figure 3(d), all the "later inserted tokens" are masked out and excluded from the previously inserted token's computation.

### 3.1.2 Efficient Computation of Offset Matrix

During training, given an *insertion order* to generate a sequence, we can pre-compute the offset matrix. However, naively constructing the offset matrix with element-wise operations takes $O(n^2)$ complexity for each sequence, limiting the training efficiency. Therefore we design an efficient algorithm to construct the offset matrix. Assume the *insertion order* is described in the form of absolute position *permutation indices* (i.e., in the previous example, [0, 6, 2, 4, 1, 3, 5]), we design a matrix algorithm called *offset compression* for our purpose. Figure 4 illustrates the algorithm with our exemplar sentence. Specifically, we first convert the absolute position vector into a matrix by duplication. Then, the upper triangular elements are masked by "infinity" to remove their impact in relative position computation because the inserting token should not attend to future to-be-inserted tokens. In the third step, each element is replaced by its in-row ranking *i.e.* its absolute position skipping the masked positions. In the final step, each row is baselined by the diagonal element to reflect that the relative position is between the last inserted token (the diagonal element) and previously inserted ones. The algorithm can be efficiently executed as a series of fast matrix operations.

### 3.1.3 An INSNET Layer

We hereby describe how to incorporate the offset matrix into each transformer layer in INSNET for efficient context encoding. In general, we inherit most designs from previous ones with relative position encoding (Dai et al., 2019; Yang et al., 2019). The encoder panel in Figure 2 (the upper left panel in purple) illustrate the process. Specifically, for layer number $i = 1...N$, given the output embedding $\mathbf{E}^{i-1}$ from the last layer and the offset matrix $\mathbf{R}$, the formulation of a transformer layer in an $N$-layer INSNET can be written as:

$$\mathbf{Q}^i, \mathbf{K}^i, \mathbf{V}^i, \mathbf{P}^i = \mathbf{W}_q^i \mathbf{E}^{i-1}, \mathbf{W}_{k,E}^i \mathbf{E}^{i-1}, \mathbf{W}_v^i \mathbf{E}^{i-1}, \mathbf{W}_{k,R}^i \mathbf{R}$$
$$\mathbf{A}^i = \mathbf{Q}^{i\top} \mathbf{K}^i + \mathbf{Q}^{i\top} \mathbf{P}^i + \mathbf{u}^{i\top} \mathbf{K}^i + \mathbf{v}^{i\top} \mathbf{P}^i$$
$$\mathbf{V}_{\text{reduced}}^i = \text{Masked-Softmax}(\mathbf{A}^i)\mathbf{V}^i$$
$$\mathbf{V}_{\text{skipconn}}^i = \mathbf{V}_{\text{reduced}}^i + \mathbf{E}^{i-1}$$
$$\mathbf{E}^i = \text{Feed-Forward}(\mathbf{V}_{\text{skipconn}}^i; \theta^i),$$

where $\mathbf{Q}^i, \mathbf{K}^i, \mathbf{V}^i, \mathbf{P}^i$ are query, key, value, position matrices, respectively, for layer $i$. $\mathbf{u}^i, \mathbf{v}^i, \mathbf{W}_q^i, \mathbf{W}_{k,E}^i, \mathbf{W}_v^i, \mathbf{W}_{k,R}^i$ and $\theta^i$ are learnable model parameters of each INSNET layer. We denote the input word embeddings as $\mathbf{E}^0$ for notation consistency. The last layer of transformer in INSNET outputs the step-wise context-aware token embeddings. They will be used to compute slot representations for the insertion steps.

## 3.2 Slot Representation and Insertion Prediction

With the context efficiently encoded, the subsequent step of an insertion-based generator is to predict the next *position and token* to insert. Thus, a representation for each potential insertion slot should be computed. We hereby show how slot representations can be *aggregated* from INSNET outputs and the potential challenges during this aggregation process.

A naive design of the slot representation is to simply concatenate the representation vectors from the left-neighbor and right-neighbor (in the natural observation order) of the slot as is done in prior works (Stern et al., 2019). This slot representation can efficiently compute the slot representation for all possible slots in parallel for each time step. However, the context encoding we obtained is insertion-order sensitive and unaware of the tokens inserted later in INSNET. This naive slot representation thus falls short of capturing the global sequence information. As a remedy, we propose to also include the last insertion token's representation to compose the slot representation $e_{i\downarrow j}^{(t)}$. Specifically, given the representations of the left-side neighbor $e_{i-}$, the right-side neighbor $e_{+j}$ and the last inserted token $e_t$, the slot representation can be computed as:

$$e_{i\downarrow j}^{(t)} = \text{LayerNorm}((f_l(e_{i-}) \oplus f_r(e_{+j})) + e_t)$$

where $f_l$ and $f_r$ are linear projections for left-point and right-point representation vectors and $\oplus$ stands for vector concatenation. The slot representation panel in Figure 2 (the bottom panel in yellow) illustrates the process. Note that comparing step 3 and step 4, both compute the slot representation for B $\downarrow$ , but the resulting slot representation changes in response to a new step due to the introduction of the global vector.

The slot representation can then be converted into a probabilistic distribution over the vocabulary to predict the log-likelihood of inserted tokens using a log-linear transformation *i.e.* $\log p(x_{i\downarrow j}^{(t)}|e_{i\downarrow j}^{(t)}) = \text{log-softmax}(\mathbf{W}_p e_{i\downarrow j}^{(t)} + b)_{x_{i\downarrow j}^{(t)}}$. To decide which slot to insert next, a position logit $\alpha_k^{(t)} = w_o^\top e_{i_k \downarrow j_k}^{(t)}$ is computed for each slot $k$ with a linear layer. Then a (log-)soft-max operation is applied on top of the position logits to obtain the log-probability to insert to each candidate slot. *i.e.* $\log o(x_{i_k \downarrow j_k}^{(t)}|\{e_\downarrow^{(t)}\}) = \text{log-softmax}([\alpha_0^{(t)}, \alpha_1^{(t)}, ...])_k$.

We apply a binary classification $q(0/1|\cdot)$ on the last inserted token's representation $e_t$ to decide the termination of the generation at each step. During training, only the final step $q(0/1|e_{n-1})$ is trained to predict 1; all intermediate steps are trained to predict 0. In the following section, we will continue to discuss how to formulate the final objective function of INSNET using these step-wise distributions.

## 3.3 Adaptive Parallelization of Insertions

In addition to our efforts for better training efficiency, we also propose an algorithm to adaptively parallelize the generation process of INSNET to speed up the decoding while preserving the generation quality. The idea mimics the graph layerization process in the Dinic's algorithm (Dinitz, 2006). Specifically, assume that we partition the tokens into different *layers*, such that the tokens within the

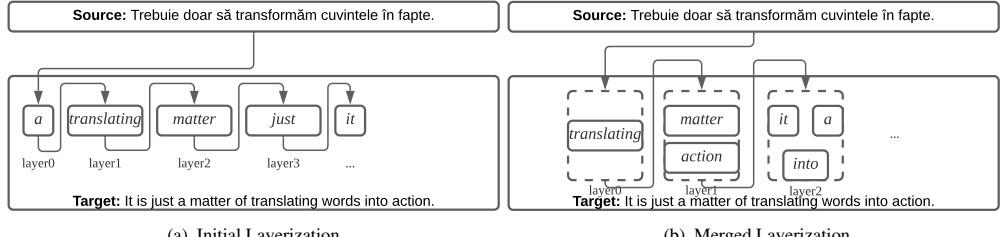

(a) Initial Layerization          (b) Merged Layerization

Figure 5: (a) Initialized layerization of tokens after the pre-training stage. It assumes a fully sequential decoding process for now. (b) As the parallelization fine-tuning proceeds, the algorithm gradually merges different layers. Tokens within the same layer are safely parallelizable in decoding without sacrificing log-likelihood over the tolerance threshold $\tau$.

same layer are *safe* to insert in parallel without affecting the coherence of context. The algorithm aims to determine the partition such that the log-likelihood estimation of the resulting parallel insertion-based model is as close to the sequential insertion-based models as possible. The intuition is that the sequential insertion process maximally captures the inter-dependencies between the output tokens. By staying close to the sequential insertion-based model regarding likelihood prediction ability, we speed up the inference without severely sacrificing the likelihood estimation quality.

We initialize the layerization with a sequential insertion order, which is equivalent to single-node layerization, as is illustrated in Figure 5(a). Starting from this initialization, we perform INSNET-Dinic to gradually evolve this fully-sequential layerization into a non-autoregressive, parallelized layerization, as shown in Figure 5(b). We *promote* a token $x_{\leftarrow}$ from its respective slot $(i^l, j^l)$ in layer $l$ to $(i^{l-1}, j^{l-1})$ in the previous layer $l-1$ if two conditions are satisfied: 1) In layer $l-1$, slot $(i^{l-1}, j^{l-1})$ does not yet have an assigned insertion; 2) $\log p(x_{\leftarrow}|\boldsymbol{e}_{i^l \downarrow j^l}^{(l)}) - \log p(x_{\leftarrow}|\boldsymbol{e}_{i^{l-1} \downarrow j^{l-1}}^{(l-1)}) \leq \tau$.

Here $\tau$ is a hyper-parameter that controls how much to parallel. Intuitively, larger $\tau$ usually indicates more tolerance for token likelihood loss caused by parallel decoding, *i.e.* more parallelization with tradeoff on likelihood estimation accuracy. The choice of $\tau$ is highly task/data-dependent as it is a value associated with the likelihood. Note that the *promotion* of tokens only affects the slot representation computation and the insertion likelihood calculation of the respective insertion. When encoding the context (as is discussed in Section 3.1), we always treat it as if these insertion operations are performed sequentially.

For a target sequence $\boldsymbol{s}_{<n} = [x_0, x_1, ..., x_{n-1}]$, after the layerization algorithm, if token $x_{i \downarrow j}$ is layerized in the $l$-th layer and to be inserted in the slot $(i, j)$, we denote it as $x_{i \downarrow j}^l$. Suppose we have $m$ effective layers in total. Denote the corresponding representation for slot $(i, j)$ given available context in layer $l$ as $\boldsymbol{e}_{i \downarrow j}^{(l)}$ and the transformer's encoding of the partial context until the end of layer $l$ as $\boldsymbol{e}_{-1}^{(l)}$. We denote the token likelihood, position likelihood and termination likelihood of the token $x_{i \downarrow j}^l$ to be $p(x_{i \downarrow j}^l | \boldsymbol{e}_{i \downarrow j}^{(l)})$, $o(x_{i \downarrow j}^l | \{\boldsymbol{e}_{\downarrow}^{(l)}\})$ and $q(0/1 | \boldsymbol{e}_{-1}^{(l)})$, respectively. The general objective of INSNET parameterized by $\phi$ can be formulated as:

$$\mathcal{L}(\boldsymbol{s}, \phi) = -\sum_{l=0}^{m-1} \sum_{x_{i \downarrow j}^l} [\log p_\phi(x_{i \downarrow j}^l | \boldsymbol{e}_{i \downarrow j}^{(l)}) + \log o_\phi(x_{i \downarrow j}^l | \{\boldsymbol{e}_{\downarrow}^{(l)}\})] - \sum_{l=0}^{m} \log q_\phi(\boldsymbol{1}(l=m) | \boldsymbol{e}_{-1}^{(l)})$$

## 4 Experiments

We demonstrate the efficiency, flexibility, and model capability of INSNET with two sets of experiments. 1) To show INSNET's performance and the flexibility to switch between sequential and parallel decoding on datasets with high inter-token dependency (i.e., less suitable for parallel decoding). We follow the setup in Zhang et al. (2020) and address the unsupervised lexically constrained text generation problem on two datasets Yelp Review and News. 2) To further verify the effectiveness of INSNET-Dinic, we evaluate INSNET as a (partially) non-autoregressive machine translation model

Table 2: Performance comparison on Yelp Review and News datasets. For Levenshtein Transformer, insertion and deletion stages are both counted in # of decoding-time iterations. It's non-trivial to do unsupervised lexically constrained text generation with auto-regressive models. To work around this, we implemented a Plan-And-Write(Yao et al., 2019) style auto-regressive transformer-based model for better reference. Models with a star mark * are re-implemented by us. Other baseline results are directly taken from the original papers.

| Model | Yelp Review | | | News | | |
| | BLEU-2/4 | NIST-2/4 | # Dec. Steps | BLEU-2/4 | NIST-2/4 | # Dec. Steps |
|---|---|---|---|---|---|---|
| Auto-regressive Transformer (Plan-And-Write-static, Yao et al. (2019)) | 16.68/5.46 | 2.79/2.86 | 39.24 | 8.79/2.40 | 1.65/1.67 | 36.74 |
| NMSTG (Welleck et al., 2019) | 10.06/1.92 | 1.11/1.12 | 27.92 | 10.67/1.58 | 2.70/2.70 | 27.85 |
| InDIGO* (Gu et al., 2019a) (w/ Searched Adaptive Order) | 16.14/4.63 | 3.08/3.10 | 45.63 | 13.89/3.62 | 3.08/3.10 | 26.78 |
| Levenshtein Transformer (Parallel Decoding, (Gu et al., 2019b)) | 14.84/3.96 | 2.84/2.89 | 14.28 | 11.76/1.89 | 2.64/2.71 | 16.13 |
| InsT-POINTER-Base (BERT init) | 11.48/2.16 | 2.15/2.15 | 6.00 | 12.13/1.63 | 2.90/2.80 | 6.00 |
| InsT-POINTER-Base (BERT init+Wiki) | 15.63/3.32 | 3.27/3.30 | 6.00 | 13.01/2.51 | 3.04/3.06 | 6.00 |
| InsT-POINTER-Large (BERT init+Wiki) | 16.78/3.79 | 3.49/3.53 | 6.00 | 14.04/3.04 | **3.28/3.30** | 6.00 |
| INSNET (Ours, Fully-Sequential) | **19.36/5.78** | **3.51/3.54** | 48.73 | **16.31/4.96** | 3.10/3.13 | 32.69 |
| INSNET-uniform (Ours) | 12.31/2.30 | 2.19/2.17 | 7.00 | 12.89/2.01 | 2.99/2.90 | 7.00 |
| INSNET-Dinic (Ours, $\tau = 10.0$) | 16.73/4.35 | 3.19/3.20 | 11.83 | 14.13/3.75 | 2.97/3.00 | 8.13 |

with parallel decoding on three classical datasets: WMT Ro-En, WMT En-De and WAT En-Ja. The general setup about how hyper-parameters are determined can be found in appendix**??**.

## 4.1 Lexically Constrained Generation

**Experimental Setup** Yelp Review dataset consists of 160K training sequences, 10K sequences for validation and 1k test sequences. News dataset consists of 268586 sentences in total, of which 10k are randomly selected as validation set, 1k for testing. YAKE [1] is performed to the test split of each dataset to extract lexical constraints.

We vary the hyperparameter $\tau$ in the range of $\{10.0, 3.0, 1.0, 0.3, -\infty(\text{fully-sequential})\}$ and collect the results on both datasets. For position prediction, we are inserting into slots with positions lying in the top 70% of the position distribution mass. For token prediction, we are doing top-$\{1, 1, 3, 3, 5\}$ sampling over the vocabulary distribution. The results are shown in Table 2, Figure 6(a) and 6(b)

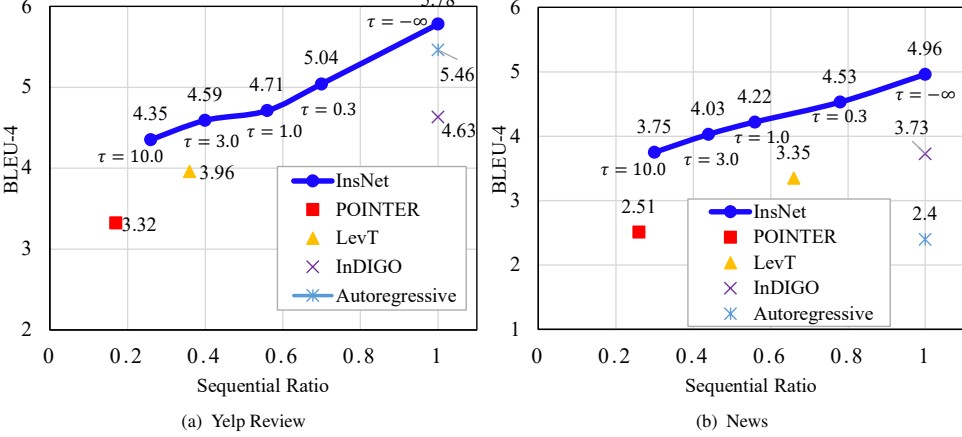

(a) Yelp Review

(b) News

Figure 6: Illustration of the spectral study results in terms of BLEU-4/Sequential Ratio in decoding on Yelp Review and News dataset. Sequential Ratio is computed by $\frac{Avg.\# \text{Dec.Steps}}{Avg.\text{Length}}$

**Discussion** 1) INSNET is able to significantly outperform most previous works on the quality-latency trade-off spectrum. With more sequential flavor (*i.e.* smaller $\tau$), it is generally to achieve the new state-of-the-art in generation quality. Compared to a Plan-And-Write style autoregressive baseline,

---

[1]https://github.com/LIAAD/yake

Table 3: Generation examples with insertion steps. Every row shows 3 consecutive steps with orange , green , blue represents the first, second, and third insertion, respectively.

| Input | day decided started focus on | local group hurt rule out |
|---|---|---|
| Step 3 | the day decided to started focus on . | the local group hurt rule out of . |
| Step 6 | the day , he decided to get started focus on . | the local group hurt the rule out of the of . |
| Step 9 | on the day , he decided to get started focus on the court . | the local group hurt the government rule out of the of the year . |
| Step 12 | but on the next day , he decided to get started to focus on the court . | the local group has hurt the government to rule out of the of the last year . |
| Step 15 | but , on the next day , he decided to get started to focus on the court for the . | the local group has been hurt the government to rule out of for the rest of the last year . |
| Step 17 | but , on the next day , he decided to get started to focus on the court for the first time . | the local group has been hurt by the government to rule out of support for the rest of the last year . |

INSNET guarantees the perfect incorporation of keywords due to its insertion-based nature. Also, its improved performance is consistent on both datasets. 2) When setting $\tau$ larger to encourage more parallelization of decoding, INSNET is mostly able to parallelize the insertions with good preservation of the generative quality. Compared to the vanilla uniformly-parallel setup (INSNET-uniform), INSNET-Dinic significantly improved the effectiveness of parallel decoding.

**Generation sample** We also show two samples of the generation process in Table 3. It is interesting to observe that the model tends to insert punctuation and function words (e.g., Articles and Prepositions) first, and then more concrete content words.

## 4.2 Non-autoregressive Machine Translation

We train INSNET-Dinic models with $\tau = 2.00$ for extended investigation of the performance on machine learning problems. The results in Table 4 show that, INSNET-Dinic is able to achieve comparable or even better performance compared to previous state-of-the-art non-autoregressive (insertion-based) machine translation models in terms of generation quality and latency.

## 4.3 Training Time Analysis

To show the training efficiency of the proposed model, we hereby do an empirical and theoretical analysis on the candidate models' training procedure. See Table 5. For empirical results, the statistics are collected on the Yelp Review unsupervised lexically constrained text generation problem. All results are collected on a single NVIDIA RTX3090 GPU. The transformer Seq2seq baseline is trained

Table 4: Machine translation evaluation. Each generation iteration of Levenshtein Transformer requires at least two full executions of the transformer model. Results with a star mark * are collected from our re-implementation. Other baseline results are directly taken from the original papers. The results for a vanilla transformer is taken from the LevT paper (Gu et al., 2019b).

| Model | Ro-En | En-De | En-Ja | #Dec Step. | Latency |
|---|---|---|---|---|---|
| InDIGO-SAO (w/o KD) (Gu et al., 2019a) | 32.47 | 26.14* | 40.87* | $n$ | 516ms |
| InDIGO-random (w/o KD) (Gu et al., 2019a) | 20.20 | 17.48* | 23.91* | $n$ | 502ms |
| InsT-uniform (+KD) (Stern et al., 2019) | 28.52 | 26.72 | 41.89* | $\Theta(\log n) \leq 10$ | 107ms |
| InsT-binary ($\tau = 0.5$, +KD) (Stern et al., 2019) | 30.66 | 27.41 | 42.17* | $\Theta(\log n) \leq 10$ | 92ms |
| LevT (+KD)(Gu et al., 2019b) | 33.26 | 27.27 | 42.36 | $\Theta(\log n) \leq 2 \times 10$ | 116ms |
| Vanilla Transformer (w/o KD) | 32.30 | 27.17 | 43.68 | $n$ | 389ms |
| INSNET-uniform | 29.13 | 26.45 | 41.67 | $\Theta(\log n) \leq 10$ | 92ms |
| INSNET-Dinic (Ours, $\tau = 2.00$) | 33.41 | 27.36 | 43.71 | $\Theta(\log n) \approx 15.8$ | 105ms |
| + KD | **33.91** | **28.05** | **44.10** | $\Theta(\log n) \approx 16.1$ | 103ms |

Table 5: Empirical & theoretical comparative study of different algorithms' training efficiency. # of Steps/Seq means during training how many time the transformer needs to be executed per sequence.

| Model | $T_{train}$/Epoch | # of Steps/Seq |
|---|---|---|
| Auto-regressive Decoder Transformer | 35min48s | 1 |
| InDIGO-SAO | 58min36s + 2h12min(SAO) | $1 + O(n)$ |
| Insertion Transformer-Sequential | 26h31min | $O(n)$ |
| Levenshtein Transformer-Sequential | 37h24min | $O(n)$ |
| Insertion Transformer | 8h24min | $\Theta(\log n)$ |
| Levenshtein Transformer | 16h33min | $\Theta(\log n)$ |
| InsT-POINTER | 9h36min | $\Theta(\log n)$ |
| INSNET/INSNET-Dinic | 1h12min | 1 |

as a Plan-And-Write model. POINTER's training time per epoch is calculated with the official implementation with mixed precision supported by the NVIDIA APEX library [2].

**Discussion**   The results in Table 5 confirm the significant improvement in training efficiency of INSNET over most previous baselines. In addition, this concretely shows that it is not practically affordable to obtain a sequential insertion-based generator directly with InsT/LevT. Note that SAO of InDIGO is an offline, inference-only algorithm that does not require any computation of gradients. Thus its constant factor is significantly smaller than other $O(n)$ gradient-propagating procedures. According to its original paper and our previous experiments, InDIGO practically needs this process to achieve reasonable performance.

### 4.4   Ablation Study on the Effectiveness of Components

Since the model components of InsNet are closely entangled, it is nontrivial to do ablation study. We design experiments to show that 1). The proposed offset matrix is a powerful insertion-oriented position encoding with significantly better capacity. We show that this particularly helps the model learn to terminate the sequence in correct timings. To study this, we truncate the offset matrix to range $[-1.0, 1.0]$ (InsNet-truncated) so that it is similar to InDIGO's position encoding, and 2) the global representation is necessary for computing the slot representation in INSNET. It in general helps to improve the sequence termination performance, and leads to convergence to better local optimum. We report the results of an ablated version of InsNet without global representation (InsNet-noglobal).

The termination NLL (see Sec 3.3 for its definition) and BLEU scores for each model can be found in appendix Table **??**. We also show some random samples from the ablated variants and the failure cases can be find in appendix Table **??**.

## 5   Conclusion & Future Work

We propose INSNET, an efficient and performant insertion-based generator that supports sequential and parallel decoding. Experiments on two unsupervised lexically constrained text generation datasets and three machine translation datasets show the advantages of INSNET over previous methods.

Future work can explore obtaining a large-scale pre-trained version of INSNET for further fine-tuning under different downstream scenarios. We anticipate such insertion-based models to have better compositional generalizability and controllability.

## Acknowledgement

We thank I-Hung Hsu, Dr. Rujun "RJ" Han, Te-Lin Wu, Kai-Wei Chang, Sarik Ghazarian, Alexander Spangher, Yining Hong, Mingyu Derek Ma and all other members from PlusLabNLP/UCLANLP group for their participation in initial discussions and comments on paper writing. We would like to thank Huggingface for their great work of the Transformers project. The work is partially supported by a Meta SRA and an Amazon research gift.

---

[2]https://github.com/dreasysnail/POINTER

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
