# A Appendix

## A.1 General Setup of Experiments

For all language generation tasks, based on whether the data is pre-processed upon release or not, either a BPE-based or a classical tokenizer is applied. Each of the evaluated transformer models, if not otherwise stated, is implemented as a *base*-sized transformer model, which has 12-layers with 12 attention head and 768 hidden dimensions. The batch size is set to 256. In cases where the model size exceeds the device capacity, the cumulative gradient trick is applied to support an equivalent optimization effect. The learning rate is selected from {5e-5, 1e-4, 2e-4}. The dropout rate is selected from {0.1, 0.2}. The weight decay rate is set to 0.02. All the models are trained with 1000 warm-up iterations and a maximum of 200000 iterations of training. A linear-decay learning rate scheduler is applied for fine-grained training of the model. If presented with a development set, the training will be early-stopped when the model stops improving for 5 consecutive epochs. For INSNET-Dinic, unless otherwise stated, the parallelization fine-tuning lasts for 5 epochs typically.

The setups of MT tasks follow previous works, including, but not limited to InsT/LevT and InDIGO (Stern et al., 2019; Gu et al., 2019b,a).

Models are implemented based on libraries like PyTorch(Paszke et al., 2019) (for the general construction of deep learning models, License: BSD) and huggingface's Transformers(Wolf et al., 2020) (for specific implementation of transformer-based models, License: Apache-2.0). Some of our baselines also used NVIDIA APEX for efficient mix-precision training [4]. We thank them for their contributions to the community. With the best of our effort we use datasets that contain either user-identity agnostic contents or those that could contain publicly reported information about famous, well-known individuals (for the News dataset).

All results that could contain randomness, if not otherwise stated, is collected from averaging the results of 3 individual runs of sampling. The variances of the results won't affect/reverse the conclusion.

## A.2 Extended Study and Qualitative Examples

Due to the page limit, we could only manage to include some qualitative examples here in the appendix. Table 6 contains some basic quantitative results for the ablated variants of InsNet. Table 7 contains a few failure cases of the ablated variants of InsNet. Table 8 contains some randomly sampled trajectories on the News dataset, showing how the context is gradually enriched by InsNet in sequentially-decoded insertion-based text generation.

Table 6: Ablation study of the position encoding and slot representation.

|  | InsNet | InsNet-truncated | InsNet-noglobal |
|---|---|---|---|
| Terminating NLL ↓ | **2.17** | 2.97 | 2.56 |
| BLEU-2/4 ↑ | **19.17/5.69** | 16.13/3.89 | 17.83/4.81 |

---

[4]https://github.com/NVIDIA/apex

Table 7: Ablated versions of the model fails to terminate the sequence at reasonable timings, leading to wordy/unreadable outputs in some cases. This is even more severe for samples from earlier checkpoints during the training.

| | |
|---|---|
| InsNet | intriguing meals and traditional . fun atmosphere . great prices . great service . |
| InsNet-truncated | ... very intriguing . great service , great staff , and the whole meals are amazing . i love the place . love the food here here . ... (**non-terminating**) |
| InsNet-noglobal | very intriguing meals . great atmosphere . and service was great . food was amazing . service was delicious . traditional italian cuisine , very fun . atmosphere is great . prices are good and very reasonable . ! . |
| InsNet | i have been here several times . great atmosphere and the service is always great . love the food , and the best steak salad . it is fantastic ! |
| InsNet-truncated | i ve been coming here multiple times in years now . great food , good food and delicious . also , the service . great atmosphere and atmosphere . service and food was great . great experience here . love food and salad , and the staff is very friendly ! . ! . |
| InsNet-noglobal | ... everything is very delicious . the whole menu was awesome , especially . we could have ever eaten since we found this place . we have had everything on the food , kids menu , , and large portions and food , great service . great place is awesome . great service , clean , fresh , tasty , tasty , hot , , , good food , service , ... (**non-terminating**) |
| InsNet | this place has been a long time to take out of my favorite spot . service is friendly and fast , the food here are excellent . fantastic sandwiches . this is my favorite spot. |
| InsNet-truncated | ... we love it here , and it s always so delicious . the service is amazing . the owner owner is always nice and has been super friendly service , and one of my favorite lunch spot . the food is always fantastic . oh and delicious . and there is the best value . the food is so fresh and the sandwiches are great . is my favorite . , it is fantastic . i .. ... (**non-terminating**) |
| InsNet-noglobal | delicious food and service . great customer service and food . good food , clean atmosphere . nice staff . good breakfast . great service , fresh food and atmosphere . will come here everyday ! . |

Table 8: Some generation trajectories collected by randomly sampling from InsNet.

| Keywords | makes election lose lost |
|---|---|
| InsNet-News | makes election lose lost |
| | trump makes election lose lost |
| | trump makes the election lose lost |
| | if trump makes the election lose lost |
| | if trump makes the election lose , lost |
| | if trump makes the election lose , lost the |
| | if trump makes the election lose , lost the election |
| | if trump makes the election lose , lost the election . |
| | if trump makes the election would lose , lost the election . |
| | if trump makes the election would lose , he lost the election . |
| | if trump makes the election would lose election , he lost the election . |
| | " if trump makes the election would lose election , he lost the election . |
| | " if trump makes the election would lose election , he lost the election . " |
| | " if trump makes the election would lose election , he once lost the election . " |
| | " if trump makes the election would not lose election , he once lost the election . " |
| | " if trump makes the election would not lose election again, he once lost the election . " |
| | " if trump makes the election would not lose election again, because he once lost the election . " |
| | " now if trump makes the election would not lose election again, because he once lost the election . " |
| | " now if trump makes the election would not lose this election again, because he once lost the election . " |
| | , " now if trump makes the election would not lose this election again, because he once lost the election . " |
| | the , " now if trump makes the election would not lose this election again, because he once lost the election . " |
| | the , " now if trump makes the election , would not lose this election again, because he once lost the election . " |
| | on the , " now if trump makes the election , would not lose this election again, because he once lost the election . " |
| | on the , " now if trump makes the election , he would not lose this election again, because he once lost the election . " |
| | he on the , " now if trump makes the election , he would not lose this election again, because he once lost the election . " |
| | he said on the , " now if trump makes the election , he would not lose this election again, because he once lost the election . " |
| | he said on the rally , " now if trump makes the election , he would not lose this election again, because he once lost the election . " |
| Golden Ref | if he makes it to the next election he ' ll lose , but the other three would have lost just as much . |

| Keywords | thought good run bring |
|---|---|
| InsNet-News | thought good run bring |
| | thought good run bring . |
| | thought good run to bring . |
| | i thought good run to bring . |
| | i thought good run to bring him . |
| | so i thought good run to bring him . |
| | so i thought would good run to bring him . |
| | so i thought it would good run to bring him . |
| | he so i thought it would good run to bring him . |
| | he had so i thought it would good run to bring him . |
| | he had a so i thought it would good run to bring him . |
| | he had a so i thought it would be good run to bring him . |
| | he had a , so i thought it would be good run to bring him . |
| | he had a , so i thought it would be good run to bring to him . |
| | but he had a , so i thought it would be good run to bring to him . |
| | but he had a great , so i thought it would be good run to bring to him . |
| | but he had a great job , so i thought it would be good run to bring to him . |
| | but he had a great job , so i thought it would be good run to bring his to him . |
| | but he had a great job in , so i thought it would be good run to bring his to him . |
| | but he had a great job in , so i thought it would be good run for to bring his to him . |
| | but he had a great job in , so i thought it would be good run for me to bring his to him . |
| | but he had a great job in , so i thought it would be good run for me to bring his son to him . |
| | but then he had a great job in , so i thought it would be good run for me to bring his son to him . |
| | but then he had a great job in , so i thought it would be a good run for me to bring his son to him . |
| | but then he had a great job in london , so i thought it would be a good run for me to bring his son to him . |
| Golden Ref | i thought , " i ' ve had a good run and if this is the way i go , bring it on . |

| Keywords | played team years completed |
|---|---|
| InsNet-News | played team three years completed . |
| | has played team three years completed . |
| | has played his team three years completed . |
| | he has played his team three years completed . |
| | but he has played his team three years completed . |
| | but he has played in his team three years completed . |
| | but he has played in his team three years before completed . |
| | but he has played in his team three years before completed the . |
| | but he has played in his team three years before being completed the . |
| | but he has played in his team for three years before being completed the . |
| | but he has played in his team for three years before being completed the year . |
| | the but he has played in his team for three years before being completed the year . |
| | in the but he has played in his team for three years before being completed the year . |
| | he in the but he has played in his team for three years before being completed the year . |
| | he in the season but he has played in his team for three years before being completed the year . |
| | he has in the season but he has played in his team for three years before being completed the year . |
| | he has not in the season but he has played in his team for three years before being completed the year . |
| | he has not in the last season but he has played in his team for three years before being completed the year . |
| | he has not been in the last season but he has played in his team for three years before being completed the year . |
| | he has not been in the last season but he has played in his team for three years before being completed the last year . |
| Golden Ref | he has not played for tottenham ' s first team since and it is now nearly two years since he completed a full premier league match for the club . |

| Keywords | local group hurt rule |
|---|---|
| InsNet-News | local group hurt rule out |
| | local group hurt rule out of |
| | local group hurt rule out of . |
| | the local group hurt rule out of . |
| | the local group hurt rule out of the . |
| | the local group hurt rule out of the of . |
| | the local group hurt the rule out of the of . |
| | the local group hurt the rule out of the of the . |
| | the local group hurt the rule out of the of the year . |
| | the local group hurt the government rule out of the of the year . |
| | the local group has hurt the government rule out of the of the year . |
| | the local group has hurt the government rule out of the of the last year . |
| | the local group has hurt the government to rule out of the of the last year . |
| | the local group has hurt the government to rule out of the rest of the last year . |
| | the local group has been hurt the government to rule out of the rest of the last year . |
| | the local group has been hurt the government to rule out of for the rest of the last year . |
| | the local group has been hurt by the government to rule out of for the rest of the last year . |
| | the local group has been hurt by the government to rule out of support for the rest of the last year . |
| Golden Ref | local media reported the group were not looking to hurt anybody , but they would not rule out violence if police tried to remove them . |

| Keywords | day decided started focus |
|---|---|
| InsNet-News | day decided started focus on |
| | day decided started focus on . |
| | the day decided started focus on . |
| | the day decided to started focus on . |
| | the day , decided to started focus on . |
| | the day , he decided to started focus on . |
| | the day , he decided to get started focus on . |
| | on the day , he decided to get started focus on . |
| | on the day , he decided to get started focus on the . |
| | on the day , he decided to get started focus on the court . |
| | but on the day , he decided to get started focus on the court . |
| | but on the day , he decided to get started to focus on the court . |
| | but on the next day , he decided to get started to focus on the court . |
| | but , on the next day , he decided to get started to focus on the court . |
| | but , on the next day , he decided to get started to focus on the court for . |
| | but , on the next day , he decided to get started to focus on the court for the . |
| | but , on the next day , he decided to get started to focus on the court for the time . |
| | but , on the next day , he decided to get started to focus on the court for the first time . |
| Golden Ref | as the day sort of went on and i decided to play , i started to focus a little bit better . |

| Keywords | continue security people privacy |
|---|---|
| InsNet-News | continue security people privacy and |
| | continue security people privacy and . |
| | we continue security people privacy and . |
| | we continue security of people privacy and . |
| | we continue security of people privacy and our . |
| | we continue the security of people privacy and our . |
| | we continue the security of people and privacy and our . |
| | we continue the security of people and privacy and our country . |
| | we continue the security of people and privacy and of our country . |
| | we continue the security of our people and privacy and of our country . |
| | we continue to the security of our people and privacy and of our country . |
| | as we continue to the security of our people and privacy and of our country . |
| | as we continue to the security of our people and privacy and safety of our country . |
| | as we continue to protect the security of our people and privacy and safety of our country . |
| | , as we continue to protect the security of our people and privacy and safety of our country . |
| | , as we continue to protect the security of our people and privacy and safety of our country , . |
| | this , as we continue to protect the security of our people and privacy and safety of our country , . |
| | we this , as we continue to protect the security of our people and privacy and safety of our country , . |
| | we do this , as we continue to protect the security of our people and privacy and safety of our country , . |
| | " we do this , as we continue to protect the security of our people and privacy and safety of our country , . |
| | " we to do this , as we continue to protect the security of our people and privacy and safety of our country , . |
| | " we need to do this , as we continue to protect the security of our people and privacy and safety of our country , . |
| | " we need to do this , as we continue to protect the security of our people and privacy and safety of our country , " . |
| | " we need to do this , as we continue to protect the security of our people and privacy and safety of our country , " said . |
| | " we need to do this , as we continue to protect the security of our people and privacy and safety of our country , " he said . |
| Golden Ref | " we ' re going to have to continue to balance our needs for security with people ' s legitimate concerns about privacy , " he said . |