# OpenReview forum: "InsNet: An Efficient, Flexible, and Performant Insertion-based Text Generation Model"
_NeurIPS.cc/2022/Conference — NeurIPS 2022 Accept_

### Official Review · Reviewer_uPbS · 2022-07-05

**Rating:** 6
**Confidence:** 3
**Soundness:** 3 good
**Presentation:** 3 good
**Contribution:** 3 good

**Summary:**

The authors propose a new insertion-based model, called INSNET, that supports both sequential and parallel decoding, to tackle the inefficient training of the naive insertion-based model. They experimented on two un-supervised lexically constrained text generation datasets and three machine translation datasets, showing that the INSNET achieves better performance in each task. The proposed approach also greatly reduces the training cost against the previous approach (as reported in Table 4).


Typos, or minor revisions for readability:

- at l. 46: "encodings , i.e." -> "encodings, i.e."

- In Table 3, the citation format will be "Gu et al." -> "(Gu et al.)" and so on

- Please re-arrange Figures/Tables display in each page. e.g.  Figure 6 could be moved to the later page so that the description will show up on the same page


+++++++++
Thank you for addressing my questions! I also checked the uploaded pdf, and I confirm the corresponding descriptions that the authors added accordingly. Yet, the manuscripts will need to be fixed a bit more for the camera-ready, so I will keep my score as "Weak Accept".

**Questions:**

- Data set up for non-autoregressive MT tasks are not found. You need to add them into Appendix accordingly.

- What kind of the error examples did you get in the experiments? Can you report any patterns if exist, or the errors that be specifically caused by the insertion-based approach?

**Strengths And Weaknesses:**

- They conducted experiments on multiple translation tasks, experimentally showing the effectiveness of the proposed approach. In Table 3, they should also report the performance using KD in other method. This seems unfair comparison. Yet, the proposed approach successfully improves the training efficiency.

- Easy to apply the proposed approach to the existing models, by introducing one pass of transformer encoding computation.

---

> ### Author Response · Authors · 2022-08-02
> **Thank you for your comments**
>
> Thank you for reviewing our paper. We really appreciate the comprehensive comments and suggestions. We have fixed all the typos and incorporated your editorial suggestions in the updated draft (uploaded). We will also conduct several more rounds of proofreading and copyediting for the final version. We respond to your other comments and questions as follows.
>
> **Performances for other methods using KD:** The reported performances of the compared baselines mostly already include KD (except for InDIGO, which is not competitive anyway). We will add additional clarification for this in the updated version.
>
> **Data set up for non-autoregressive MT tasks:** We simply follow the common setups of these MT datasets as previously existing models. We have updated the appendix to include the information (See Appendix A.1).
>
> **Error examples or patterns**: Thank you for the suggestion. We will conduct an error analysis for the camera-ready version. For now, we have added some qualitative examples of the generation outputs from our model to Appendix Table 5 in the updated draft (uploaded).

---

### Official Review · Reviewer_EEGj · 2022-07-06

**Rating:** 4
**Confidence:** 3
**Soundness:** 3 good
**Presentation:** 1 poor
**Contribution:** 3 good

**Summary:**

The paper proposes InsNet, an insertainn-based text generation model. It introduces a new insertion-oriented relative position schema for efficient encoding and computation. It introduced an algorithm to adaptively parallelize the generation process to speed up the inference. The paper conducts comprehensive experiments on a few text generation tasks, and the result is competitive.


**Questions:**

What’s KD?
I can't fully understand Sec 3.2 and 3.3.

**Limitations:**

Poor writing and presentation.

**Strengths And Weaknesses:**

The paper introduced “offset”, an insertion-oriented relative position encoding schema. It overcomes the problem of prior methods and reduces the re-encoding problem. It works well in practice. The paper also offers an algorithm which decomposes this method into a few fast matrix operations. It’s a novel algorithm which can be widely applied on future insertion based models.

The paper rigorously and closely compares with prior work. The proposed method works well on most datasets. The results are convincing.

Weakness
The paper is very hard to follow. I can’t quite understand Sec 3.2 and 3.3, after spending 1+ hours reading this paper and some of the prior work (InsT and InDIGO). There are many concepts and terminologies in this paper that I can’t fully understand. eg. Dinic’s algorithm, fully-sequential layerization, tau, etc. I even had a hard time to understand the offset algorithm. I am not sure how the model works as a whole system (input, output, modules, training, inference, etc.) I was very excited about the paper, but I am frustrated after spending 2 hours.
I would give some suggestions:
1. Use a simpler example. “<s> I have a pen . </s>” is too long for demonstrating the algorithm. It’s such a pain to read Figure 2, 3, and 4. I believe a sequence like ABCD is visually better and easier to remember.
2. Elaborate the core equations. There are a good number of equations in Section 3, but it requires decent patience and memory to encode, understand and remember them. Please focus on core/your equations, and illustrate them with figures and proper notation.
3. Move something to appendix. The paper gets pretty crowded and complicated. You can also consider just get rid of some components.
4. Show some example outputs. I am curious about how it empirically works and what’s the order of generation.

---

> ### Author Response · Authors · 2022-08-02
> **Thank you for your feedback and some clarification**
>
> Thank you for your effort in reviewing our paper. We certainly understand the frustration of not being able to understand some technical details of a paper. However, we clarify that our paper is not the type that only has a small $\delta$ improvement over an existing work. Therefore, it naturally requires more time to read and understand the technical details.
>
> We want to reiterate our major contributions: we came up with a completely different design of insertion-based generation models to support efficient training and parallel decoding. Our major innovations include the position encoding, the slot representation, and Dinic's algorithm to automatically decide how much to parallelize for the insertion operations. We believe our contributions are significant and solid, and our experiments support our claim. We would really appreciate it if you could spend more time reading and re-evaluate our paper.
>
> To answer your question: we have added some further explanations of the intuition for $\tau$ in the general response. KD means knowledge distillation, which is commonly used in non-autoregressive or even some of the SOTA autoregressive/seq2seq machine translation models.

---

> ### Author Response · Authors · 2022-08-09
> **We added an illustrative figure and some qualitative examples in the appendix**
>
> Dear reviewer,
>
> We just added an illustrative graph for the model overview with illustrations of each component (position encoding, context encoder, slot representation, and the adaptive parallelization algorithm) and some generation samples to the appendix. We hope they help demonstrate the model better and provide intuitive ideas about the model's generation. Please check them out. Thanks!

---

### Official Review · Reviewer_CKci · 2022-07-11

**Rating:** 5
**Confidence:** 3
**Soundness:** 3 good
**Presentation:** 2 fair
**Contribution:** 2 fair

**Summary:**

This article proposed INSNET, an insertion-based text generation model, for efficient training and flexible decoding. Specifically, a new insertion-oriented position encoding method is used in INSNET to enable efficient computation and a hyper-parameter is provided in INSNET to flexibly control the parallel degree. Experiments were performed to demonstrate INSNET’S advantages in terms of training speed, inference efficiency, and generation quality.

**Questions:**

1. What is the precondition to use this position encoding method, this method seems heavily rely on device support (eg GPU support on matrix operation). Under the precondition, what extra cost would it cause considering it is time-efficient, is it memory efficient compared to other methods?

2. From what Figure 3 illustrated, it seems that the algorithm actually recomputes position encoding in each step, thinking that in each time step, the sequence position is overall recomputed throw matrix operation. In fact, there is no need to recompute previous token position encoding. So, is there any way to go further to avoid this computational waste and maintain efficiency in the same way?

3. t is an important hyper-parameter for INSNET since it can control how much to parallel. It makes sure the flexibility of the model. Experiments should be made to demonstrate that truly influence the parallelism of the model. What is the relationship between and Sequential Ratio?  Are there any formulas that could be given to show how it manipulates the parallel capability of the model?

4. The inference time comparison could also be analyzed to show the model’s efficiency. More analysis about inference time comparison should be given.

5. Compared to the traditional transformer, the training speed is slower, why happen? There should be some reason to be listed in the paper and verified?

6. Since the non-autoregressive transformer is not that common, when introducing INSNET, a figure about it could be given. More explanation could be given to it.

7. Figure 6 shows the performance of INSNET under different Sequential Ratio, could the same experiments be executed for another model to show the overall performance comparison among these models under different Sequential Ratio?

8. The definition of the Loss function contains an error. In the second part of the formula, the symbol l is from 0 to m, while in the first part, l is between 0 and m-1. They are not consistent.

**Ethics Review Area:**

["I don’t know"]

**Strengths And Weaknesses:**

Strengths:

This article proposed a new insertion-based text generation model, which is efficient in time consumption and flexibly supports parallel and sequential decoding.

Weaknesses:

There lack ablation experiments to verify the effect of three major components of INSNET.

It is unclear what useful information is captured by one pass of context encoding in INSNET.

---

> ### Author Response · Authors · 2022-08-02
> **Thank you for your review & some clarification**
>
> Thank you for the detailed comprehensive comments and suggestions. We want to clarify some misunderstandings through answers to your questions, and we hope you can reconsider your rating after we clarify some questions/doubts. Thanks!
>
> We respond to your request for an ablation study in the general response. Specifically, we have included some ablation studies in the updated draft in Appendix Table 5, and will consider adding more for the camera-ready version.
>
> **Precondition and cost to use the proposed position encoding method:** We clarify that the acceleration of the proposed algorithm requires exactly the same hardware conditions as what common deep neural networks and the vanilla transformer require. The conversion from insertion order (permutation ids) to the final position encoding (offset matrix) does not involve any gradient calculation and only requires a matrix of size LxL for a sequence of length L. This requires no more memory/computation than existing models like Transformer-XL or XLNet.
>
> **Redundant computations for the position encoding:** We clarify that we did not recompute position encoding in each step, and we have already optimized our algorithm to compute the offset matrix for each instance only once throughout the training. This is where the acceleration of the algorithm comes from, compared to context re-encoding methods like Insertion Transformer/Levenshtein Transformer.
>
> **The relation between $\tau$ and the sequential ratio**: The relation itself is highly data-dependent and we are afraid that there's no closed-form solution for it. However, for the same data, higher $\tau$ encourages more parallelization and lower $\tau$ reduces the generated conflicts. The intuitive/logical connection between the two is described in Sec 4.1 and illustrated in Figure 6. We will update the illustration for the corresponding experiments (Figure 6) to better plot such a connection between $\tau$ and the Sequential Ratio.
>
> **Inference time comparisons/analysis**: The respective analysis is discussed in Sec 2. Each class of insertion models has its characteristic inference time O-complexity, but with different constant factors which could only be clearly shown in the actual run time. In general, during inference(decoding), models with parallel decoding require O(\Theta log n) complexity and sequential models require O(n).
>
> **Why the training is slower than traditional transformer**: The traditional transformer uses absolute positions which is computationally faster than most implementations of _relative positions_, including ours.
>
> **An illustrative figure for InsNet**: We appreciate this suggestion and highly agree with it. However, given the page limit. We find it hard to include such an illustration. We can include one if we have one additional page for the camera-ready version.
>
> **The same experiments of performances under different Sequential Ratio for another model**: Unfortunately according to our knowledge InsNet is the first and only model that can achieve such flexibility in parallelized insertions (with a tractable training/inference complexity).
>
> **A typo/mistake in our formulation**: This is not a mistake. The first term computes the likelihood of each insertion operation (position likelihood + token likelihood). For a sequence of m tokens, we have m sub-terms in the first part of the loss function. The second term computes the termination likelihood. For a sequence of length m, it is only terminated after the last (m-th) token is already part of the generated token. Thus, we will have m + 1 termination likelihood sub-terms.

---

> > ### Author Response · Authors · 2022-08-09
> > **We added an illustrative graph in the appendix**
> >
> > Dear reviewer,
> >
> > Thank you for your acknowledgment of reading our rebuttal. I hope it addressed at least some of your concerns.
> > We just added an illustrative graph for the model overview in the appendix and will move it to the main text if the paper is accepted and we have an extra page. I hope it helps explain the model. Please check it out. Thanks! We really appreciate your time!

---

### Official Review · Reviewer_iCQn · 2022-07-12

**Rating:** 7
**Confidence:** 4
**Soundness:** 3 good
**Presentation:** 3 good
**Contribution:** 3 good

**Summary:**

The authors propose an efficient insertion-based text generator (INSNET) that does not require re-encoding of the context. The model follows the insight to enable computation sharing and aggregation. The proposed model is also flexible for the controlled generation and parallel decoding. Experiments on two unsupervised lexically constrained text generation and machine translation tasks verify the model.

**Questions:**

Is it possible to conduct an ablation study with only position representation or slot representation? How much impact does each of these elements contribute?

**Strengths And Weaknesses:**

strengths:

1. The proposed model is versatile and intriguing, which contributes to general text generation models. It also achieves a better quality-latency trade-off compared to former edit-based generations.
2. The novel and well-motivated scheme for position representation and insertion slot representation, which is insightful for the text generation community.
3. The efficient computation sharing strategy is well-illustrated. The adaptive training scheme is easy to understand as it follows the curriculum learning to start from generic decoding to arbitrary insertion orders. Though it brings certain overhead, experiments show the decoder stays efficient.

weaknesses:

1. Relatively lack of experiments for analysis.
2. The impact of hyper-parameter $\tau$ is task-specific and may be costly to tune. The paper should provide some quantitative analysis, e.g., why machine translation can achieve decent quality with smaller $\tau$.

---

> ### Author Response · Authors · 2022-08-02
> **Thank you for the useful feedback**
>
> Thank you for your time and useful feedback. We will follow the comments to further polish our paper. We provide detailed responses to your requests for having an ablation study and further investigating the impact of $\tau$ in the general response. In particular, we have added some ablation studies in the updated draft in Appendix Table 5.

---

### Author Response · Authors · 2022-08-02
**General Response to The Ablation Study and The Hyperparameter $\tau$**

**Ablation study**
We appreciate the feedback from reviewers requesting ablation studies of the model components. We clarify that the proposed components for InsNet are integrated and entangled with each other to support efficient training and parallel decoding. It is hard "ablate" a component. However, we followed reviewers' feedback and tried our best to come up with some inferior variations for our position encoding and slot representation computation. As expected, the ablated versions perform worse than our full model. We have updated our draft to include some primitive results. Results on the Yelp review dataset (an unsupervised lexically constrained text generation problem) are updated in the after-review revision. See Appendix A.2.

**Intuitive Understanding of $\tau$**
In Sec 3.3 we discussed that larger $\tau$ encourages more parallelization of insertions. However, the absolute value of $\tau$ is highly task-dependent because it is a value associated with likelihood. Therefore, it is hard to compare the value of $\tau$ across different tasks/datasets. For machine translation, most information in the output sequence is dictated by the input (i.e. the likelihood of each insertion operation is less dependent on the generated contexts given the input). We ended up having small $\tau$ for the MT datasets because small $\tau$ for those datasets already encouraged decent parallelization. We will include more statistics in the camera-ready version.

**Further Investigation of $\tau$**
We actually have shown the results with annealed $\tau$ s in our LCG experiments. For MT tasks, since the respective study can be computationally expensive, we won’t be able to present them in such a short amount of time in the after-review revision. We will try to conduct a larger range of studies also for MT datasets in the camera-ready version.

---

### Author Response · Authors · 2022-08-05
**Looking Forward to Further Discussion**

Dear Reviewer(s),

We truly appreciate your comments and feedback from the first round of review and we hope our primitive results/responses for addressing them can be reasonable factors for you to consolidate/reconsider your evaluations. As the end of the author-reviewer discussion period draws near, we are wondering if there should be any follow-up questions regarding our responses that you'd be willing to discuss. We would really appreciate it, thanks!

---

### Meta-Review · Area_Chair_Biks · 2022-08-29

**Recommendation:** Accept
**Confidence:** Certain

**Metareview:**

There is consensus between 3/4 reviewers regarding acceptance and the review that suggests borderline rejection also admits the merit of the paper.  Given the new proposed method for insertion based generation in transformers and the experimental results, I think the paper should be accepted to Neurips.

**Award:**

No

---

### Decision · Program_Chairs · 2022-09-14

Accept